# How to Achieve a Continuous Increase in Public Transport Ridership?—A Case Study of Braunschweig and Tampere

**Christoph Schütze [1], Nina Schmidt [1], Heikki Liimatainen [2] and Thomas Siefer [1,\*]**

[1] Institute of Transport, Railway Construction and Operation, Department of Architecture, Civil Engineering and Environmental Sciences, Technische Universität Braunschweig, 38106 Braunschweig, Germany; c.schuetze@tu-braunschweig.de (C.S.); nina.schmidt@tu-braunschweig.de (N.S.)

[2] Transport Research Centre Verne, Tampere University, 33014 Tampere, Finland; heikki.liimatainen@tuni.fi

[\*] Correspondence: th.siefer@tu-braunschweig.de; Tel.: +49-531-391-63600

**Abstract:** This manuscript is based on cooperation between the universities in Tampere, Finland, and Braunschweig, Germany. One of the main goals of the two institutes involved in research for public transport is a continuous increase in ridership. Therefore, the aim of this study is to (1) investigate the level of service attributes of public transport in Tampere and Braunschweig and evaluate their current state and (2) give advice to each city to further increase the ridership. At the beginning, the individual attributes and indicators for comparing public transport in both cities are presented and combined with theses from the literature. The attributes are divided into four chapters: "Level of Service", "Pricing", "Information and Marketing" and "Quality of travel". In the following, the individual indicators, such as the travel speed (Level of Service) or information at stop (Information and Marketing) of the two study areas, are analysed and evaluated. Additionally, the public transport system in both investigated cities is expanded with new a tramway system (Tampere) or new tramway expansions (Braunschweig), which gives a lot of opportunities for improving the attractiveness of public transport.

**Keywords:** public transport; case study; continuous increase; comparison; rural areas; bus; tramway

## 1. Introduction

Mobility is a basic need that everyone should have access to and that is part of an attractive standard of living. Today, people are travelling longer distances at higher speeds than ever before. In Germany, around 40 km are travelled per inhabitant every day. These distances are usually covered by car [1]. This traffic causes high emissions of carbon dioxide ($CO_2$), noise, and traffic jams every day [2]. Some cities have recognised the negative effects of today's mobility and lifestyle and are actively taking steps towards a more sustainable level. Providing mobility through a well-developed and reliable public transport system is particularly important to mobilise people to switch from their own cars to public transport [3].

This research presents a case study of two cities which have successfully increased public transport ridership during the last 20 years. The increase in public transport users in these cities is based on population growth and adjustments in public transport. The aim of this study is to (1) explore the level of service attributes of public transport in Tampere and Braunschweig and evaluate their current state and (2) give advice to each city to further increase their ridership.

This study focuses on the public transport systems in the cities of Braunschweig and Tampere. These cities were selected because they are of similar size in terms of population and have a similar

modal share of public transport. Both cities have also been successful in increasing public transport ridership and are developing tramway systems, with Tampere currently constructing the first tramway system to be opened in 2021 and Braunschweig with an extensive current tramway system and plans for further developments.

## 2. Theoretical Background

Mobility has become an integral part of everyday life. Nowadays, people are travelling longer distances in less time. The number and duration of trips has not changed over the last 50 years overall. Nowadays, while much longer distances are covered for the same things, the effort remains the same. Therefore, it is important that longer distances can be covered by public transport in the same time in order to increase the acceptance of public transport. A change must take place so that spatially and temporally separated activities such as living, working, and leisure activities can be combined with each other. About 40 km and 3.1 trips per inhabitant are travelled every day in Germany. This takes an average of one hour and 20 min per person [1]. In Finland, people travel on average 41 km, 73 min, and 2.7 trips per day [4]. Despite an increase in public transport passengers, the car remains the dominant mode of transport. A total of 57% of all journeys and 75% of all passenger kilometres are covered by car in Germany [1], while in Finland the shares are 59% and 76%, respectively [4]. There are 43 million cars in Germany, which is an average of 1.1 cars per household [1], while in Finland there are 2.7 million cars, which is an average of 1.0 cars per household [5].

For the theoretical background, it is necessary to analyse the customer's point of view and their limits of acceptance for the choice of means of transport. In the following, the state of research is described by discussing knowledge from the field of transport research by outlining the qualitative factors, the price elasticity, as well as the travel time dependency. Decisive criteria for the use of public transport are prices and the range of tickets, travel speed, reliability, comfort, flexibility, accessibility, and safety [6].

### 2.1. Quality of Service Attributes

A continuous increase in public transport ridership is a goal affected by multiple factors leading to customer satisfaction and loyalty. In Finland, the "Act on public transport services" (869/2009) requires public transport authorities to define the quality of public transport services in the area. In Tampere, the following attributes have been determined in the latest document, with hours of operation, frequency, and walking distance to the nearest stop as quantitative attributes, and with reliability, ticketing system, clarity of the route network, infrastructure, travel time, transfers, and vehicles as qualitative attributes. There are six levels of quality of service with determined aims for each level—for example, "high modal share of public transport" is on level 6 (highest) [7].

There are numerous studies on the attributes of the quality of public transport using varying phrasing of the attributes, but the views on the key attributes seem to converge under the attributes presented in Table 1.

**Table 1.** Public transport quality of service attributes in the literature.

| Quality of Service at Tribute | Accessibility/ Ease of Use | Frequency | Speed/ Travel Time | Reliability/ Punctuality | Information | Courtesy/ Staff Behavior | Safety | Security | Comfort | Price/ Value for Money | Social Image |
|---|---|---|---|---|---|---|---|---|---|---|---|
| BEST [8] | | x | | x | x | x | x | x | x | x | x |
| Walker [9] | x | x | x | x | x | x | x | x | x | x | |
| Mouwen [10] | x | x | x | x | x | x | x | x | x | x | |
| Redman et al. [11] | x | x | x | x | x | | x | x | x | x | x |
| Tyrinopoulos & Antoniou [12] | x | x | | x | x | x | x | x | x | x | |
| de Ona et al. [13] | x | x | x | x | x | x | x | | x | x | |
| dell'Olio et al. [14] | | x | x | | | x | | | x | | |
| Guirao et al. [15] | x | x | x | x | x | x | x | x | x | | |
| Abenoza et al. [16] | x | x | x | | x | x | | x | x | | |

The quality of service attributes includes both objective and subjective factors, thus enabling comparison between cities directly using objective factors and indirectly through customer surveys using subjective factors. Comparisons are made, e.g., between large European cities regularly within the Benchmarking in European Service of Public Transport [8] network using customer surveys. Based on the BEST (Benchmarking in European Service of Public Transport) survey results, there are differences in the importance of the factors, but the following issues belonged to the five most important factors in all cities: number of departures and easy transfers. In addition, the following was mentioned in all but one city: public transport mostly runs on schedule. Punctuality and frequency were found to be the most important attributes also by de Ona et al. [13], dell'Olio et al. [14], and Guirao et al. [15]. Each public transport user assesses the quality components differently. The interaction between the different factors is not determinable yet. The public transport system can be influenced by further disturbing factors [17]. Decisive criteria for the use of public transport are prices and the range of tickets, travel speed, reliability, comfort, flexibility, accessibility and safety [6].

The study by Redman et al. concludes that improvements in the quality of service of public transport have the potential to attract private car users. The question of what improvements exactly should be made depends primarily on individual motivations for using private cars. In some cases, improving accessibility may be sufficient, but in other cases it may not. For some users, less crowding, improved safety, cleaner stations, and better user information are more important than physical attributes such as speed.

Motorised individual transport (MIT) should not be abolished completely, but the public transport system should be changed in a sustainable way so that it makes people rethink and switch to public transport. Comfort, quality of life, and the environment should not suffer during the change [11]. In accordance with the attributes found in the literature, the following attributes and indicators are used to compare the public transport systems in Tampere and Braunschweig. Four different categories and their corresponding attributes are displayed in Figure 1. Those attributes are used because they contain a broad range of aspects to evaluate and compare the attractiveness of public transport systems. In the following figure, each attribute is explained shortly regarding its category.

| Level of Service | Pricing | Information&Marketing | Quality of travel |
|---|---|---|---|
| **On time performance** „Accuracy of the realized depature times in relation to the schedule" | **Price of tickets** „Price of various types of ticket and season cards" | **On board information** „Static, dynamic and vocal information on delays etc." | **Ease of boarding** „Ease of boarding and alighting from the vehicle" |
| **Travel speed** „Average travel time including transfer, travel and stop times for used transportation modes" | **Ticket selling network** „Ease of obtaining a ticket from on- and off-board selling points" | **Digital information** „Digital information on delays etc. via app or browser" | **Seating capacity** „Ratio of seats required to seats available" |
| **Service frequency** „Interval of travel services offered by transportation mode and differentiated after day and night time" | **Complexity of fare system** „Interval of travel services offered by transportation mode and differentiated after day and night time" | **Information at stops** „Analog information about travel times, stops and lines at stops" | **Safety and Equipement at stops** „Safety and equipement at stops and terminals for waiting passengers" |
| | | **Brand appearance** „Company appearance and attractiveness for customers" | |

**Figure 1.** Attributes and indicators to compare public transport based on Mouwen [10].

*2.2. Level of Service*

The accuracy of the realised departure time in relation to the schedule is the most important criteria for public transport users. They rely on punctual services to either get to their final destination on time or to be able to use planned transfer connections. The on-time performance is the more important the lower the service frequency. For example, the on-time performance in a metro system with a three-minute interval does not matter to public transport users. The reliability of public transport is an equally important criterion for making local transport more attractive. Reliability becomes

particularly important when a changeover is necessary on the respective route. Due to delayed public transport systems, transfer connections are often not achieved. If a transfer relationship is not achieved, the travel time extends enormously and public transport becomes less attractive for its users. The lack of information about delays is also perceived as very annoying for users. Information systems must therefore be reliable with real-time data in order to increase user acceptance. If the frequency of public transport systems is too low, it is not possible to use public transport without detailed planning [6].

The second attribute for the level of service is the travel speed, which includes the way to and from the bus/train stop, as well as the travel time itself and transfer between different services. The speed of public transport systems may vary by the maximum average speed, independency of their infrastructure, and frequency of stops. The slowest public transport system is the city bus. The tramway system is designed with an average speed of between 20 and 25 km/h [18].

Additionally, the time necessary to enter the public transport systems needs to be taken into account. A distance of more than 10 min to a nearby stop by foot is in most cases unacceptable for potential public transport users, while a 5 min walking distance is usually accepted. Many potential customers do not even tolerate a change in vehicles. With more than two transfers, the acceptance decreases for almost all users [17].

A distinction is made between comfort and effort, whereby the effort refers to the problem of the entries and exits as well as transfer connections in public transport. Private motor vehicles (PMV) have only two of those interfaces (from the starting point to the car, the PMV journey, and from the parked vehicle to the destination) [17]. Especially rural areas are affected, because of the relatively long distances between public transport stops. Furthermore, the travel time is reduced because of fewer stations on the way downtown.

Another attribute for the level of service is the service frequency, which normally depends on the transportation mode, density of the surrounding area, and time of day. The frequency is defined by user demand, vehicle capacity, and economical boundaries. Overlaying lines at stops can reduce the interval of public transport services for users [18].

### 2.3. Pricing

Travel expenses are defined as financial expenses resulting from geographical shifts. Costs for public transport are mostly charged via ticket [17]. Costs for the maintenance of and investment in a private vehicle are often not considered in a price comparison with public transport [19]. In 2017, the Allgemeine Deutsche Automobil Club e.V. (ADAC, automobile club) conducted a survey in 10 major German cities on the subject of the non-use of public transport. In this survey, 62% of respondents replied that the ticket price is too expensive for public transport. The costs of public transport are the second biggest obstacle, besides the general preference for cars [6].

Besides the price of the ticket, the way to buy a ticket and its complexity are important aspects for public transport users. A too complex range of tickets is also mentioned as a reason for not using public transport [6]. There are different ways of selling tickets for public transport companies, for example from the driver or online via an app. There are a lot of single ticket users in public transport systems and the way to buy a ticket influences comfort, but figuring out which ticket is the correct—or better, the cheapest—choice is another issue and poses a barrier for new users. Therefore, the complexity of the two fare systems and the ticket selling network in Tampere and Braunschweig will be investigated.

### 2.4. Information and Marketing

Passenger information is a key element of communication within public transport marketing. The function of passenger information is to act as an operating manual for public transport use. It is intended to create trust, provide orientation, and encourage potential users to use public transport. Passenger information should be understood as a key component of the accessibility of public transport [20]. The use of means of transport is becoming more flexible and individual. The means of transport must be better connected to each other in order to make public transport more attractive.

The barriers to access multimodal and intermodal services are still too high for most customers and the benefits too low. Thus, mobility platforms need to be better developed. The mobility platforms are based on information systems that display user-friendly mono- and intermodal connections. Neutrality and non-discrimination in the presentation of connection results must be ensured if the offer is to be successful [21].

Therefore, the current status of onboard, online, and stop information will be investigated for Tampere and Braunschweig. Besides the information about on time performance, departures, ticket prices, fare systems, etc., the brand appearance is a big issue for marketing opportunities.

*2.5. Quality of Travel*

Other than effort and comfort, a quality criterion to increase the number of passengers in public transport is accessibility. Not only people with disabilities are affected, but also people with strollers or heavy luggage. In Germany, the "Act to Amend Regulations on the Transport of Persons" is intended to achieve complete accessibility in public passenger transport nationwide by the beginning of the year 2022. In Finland, there is no similar legislation, and the accessibility for special groups is addressed in separate legislations. According to the ADAC survey, 60% of respondents say that public transport is unsuitable for transporting goods and 51% say that the vehicles are overloaded [6]. Therefore, the ease of boarding is investigated.

Another important aspect is the travel comfort. Public transport vehicles are often crowded, especially during rush hours, and it is therefore not comfortable for many people to ride a crowded bus if the alternative is their private car. That is why the seating capacity of public transport vehicles is an important attribute to evaluate its attractiveness. Another aspect of quality of travel is safety, which can be differentiated into safety at stops including equipment and the safety of public transport users while traveling. While the safety aspect of the MIT is primarily about traffic safety, the safety aspect of public transport is about ensuring that vehicles and stops are safe for users even during night and off-peak hours [6]. Public transport stops will therefore be evaluated for both cities.

The traffic safety of public transport is significantly higher than that of MIT. An association of interests for railways evaluated the death and injury statistics from 2007 to 2016. The risk of death for car users was 53 times higher than for rail passengers and 12 times higher than for bus passenger, and the probability of injury was 125 times higher respective 3 times higher [22].

## 3. Methods Used

Different methods are used in this study to evaluate the cities' transport networks and to determine factors for increasing the public transport ridership. First, a deep on-site analysis was necessary for gathering data to be able to understand and describe the status quo of the local public transport in Tampere and Braunschweig. This particularly includes the usage of the public transport network over a period of time and interviews with responsible authorities of different companies involved in local transport in both cities.

These two actions provide, inter alia, information for:

- A local transport plan with a general overview (lines, schedule etc.);
- Inside looks into the fare model used;
- Information about the condition of vehicles and stops which affect the passenger comfort.

As a status quo, the year 2018 was determined. There will be a lot of changes and developments both in Tampere and Braunschweig in the years ahead, especially in expansion projects for tramways. In chapter 4, the two public transport systems are described in their current state. Based on the gathered information, the public transport systems in both cities are analysed by qualitative factors, which were described in chapter 2 for the choice of the means of transport. Fulfilling those qualitative factors to a certain level can be evaluated as a success factor for public transport. The evaluation is performed on a

qualitative basis, taking all information collected into account. That information can help to explain the rise in ridership, along with some other factors such as network expansions or increased population.

Neither Braunschweig's nor Tampere's public transport system is near perfect, hence there are things to improve that are derived on the basis of qualitative factors. Additionally, there are some lessons that the cities' public transport systems can learn from each other. The authors evaluate the gathered information and use expert knowledge to derive results.

## 4. Public Transport Systems in Tampere and Braunschweig

In this chapter, a short introduction for the two public transport systems in Tampere and Braunschweig is given. For Tampere, only bus and rail transport are available in the year 2018. The construction of a new tramway system is already ongoing but not finished. Because of its high value for the whole public transport system in Tampere, the new tramway is introduced with a brief description. In Braunschweig, there are bus, rail, and tramway transportation systems in use for public transport. In addition, there is also an extensive tramway expansion in the planning stage. In both cities, there are different sharing concepts, which include bicycles and cars and will be briefly described later in this chapter.

### 4.1. Traditional Public Transport Systems

Tampere has a well-developed bus network with 59 lines in winter, which are extended for tourism in summer. A total of 18 million km per year are covered by eight bus companies. There is a higher frequency in the city centre than in the suburbs. According to the director of public transport, the internal operator served 43% of the network in 2017. Regional transport is organised according to the purchase provider model. While the purchasing plans; the organisation of public transport; and the fares, tariffs, and marketing are carried out by the public transport authority Nysse, the service providers assume responsibility for the operation [23].

In spite of the area-wide bus connections, the number of PMV in the city of Tampere is increasing. This is a result of the growing population, which leads to the streets reaching their limits. To serve the increasing amount of people, it is necessary to expand the public transport by developing a light rail/tramway system [23]. The operation of the system with two routes is going to be built in two phases and it is planned to start operating in 2021. During the first phase, a 15 km-long track between the hospital TAYS and Hervanta will be constructed. The second phase consists of a 6.6 km-long track to the northwestern part of Tampere [24]. At the end of the route, there is the possibility to transfer from the tramway to the bus. The new tramway system is not discussed in the following analysis of current state, as it is not yet operational.

The tramway in Braunschweig exists since 1879 with a gauge of 1100 mm. Today, there are five lines with a total length of 46.2 km. The travel times are very different, ranging from 16 min (line 4) to 45 min (line 1). A total of 81% of the current network is on segregated or independent track. Therefore, tramway traffic is less dependent on road traffic and can be prioritised at the traffic lights. This leads to a less delayed operation and thus to a higher acceptance of the passengers [25]. In 2017, the tramway had 19.3 million passengers and 47 million passenger kilometres [26]. To develop sustainable transport, the authorities in Braunschweig are planning to expand the tramway. There are different lines projected, with a maximum expected cost of 208 Mio. Euro [27].

Another important means of transport in Braunschweig is the bus. The Braunschweiger Verkehrs-GmbH (BSVG) operates 38 bus lines in the city [28]. Most of the bus lines are radial and cross-city links. Besides this, there are some feeder lines connecting districts with tramway stops. In 2017, the bus lines covered 155 million passenger kilometres and carried 20.5 million passengers [26].

### 4.2. Sharing Mobility Concepts

In Tampere, there are two car-sharing providers and one bike-sharing provider. BloxCar is an operator that specialises in peer-to-peer PMV sharing [29]. 24Rent, on the other hand, operates its own

fleet of vehicles for rental [30]. In summer 2017, a station-based bike-sharing project called EASYBIKE was introduced in Hervanta [31]. It is based on the Hervanta campus of the Tampere University.

In Braunschweig, there are two car sharing providers, Flinkster and Greenwheels. Both are station-based and have access to more than 40 vehicles [32,33]. The bike-sharing operator in Braunschweig is "Call a Bike" from Deutsche Bahn [34].

## 5. Analysis Based on the Level of Service Attributes

In this chapter, qualitative factors for the public transport systems in Braunschweig and Tampere are analysed. Both cities experienced a rise in lines and changes in frequencies and schedules, but the main focus is on the qualitative factors, which are described in chapter 2. These factors can be differentiated into travel costs, travel time, and travel comfort. The aim of this analysis is to identify possible improvements to improve the attractiveness of the public transport systems in the two investigated cites. These improvements are given in chapter 6.

The people in Tampere/Finland have a more active tendency towards continuous improvement by taking up innovations and integrating them into the public transport than in Germany. For example, the Mobility as a Service (MaaS) concept has its roots in Helsinki and has been implemented in Finland for many years. In Tampere, smartcards and QR codes have been used as public transport tickets since 2000. In recent years, an increase in the number of passengers has already been identified, largely through the creation of a uniform tariff within the association and a reduction in prices for some tariffs. Despite all these measures, further improvements can continue this trend and lead to a further increase in the number of passengers.

### 5.1. Level of Service

#### 5.1.1. On-Time Performance

Travel times for passengers should be as short as possible, as described in chapter 2. The type of traffic routing and a possible priority for public transport can strongly influence the travel time and on-time performance.

In Tampere, buses are equipped with GNSS receivers and 3G/4G connections, which provide the real-time information of the locations of every bus in the fleet, and the data is shared openly through the Journeys API [35]. Real-time punctuality data are provided through journey planners on websites, apps and at stops.

In Braunschweig, BSVG does not record any punctuality data for tramways and buses. This is expected to change in 2020, through switching from analogue radio to GSM [36]. On most of the network, tramway lines are separated from the normal road traffic. At crossings, it is possible to prioritise tramways, as they have their own traffic light system [37]. Segregated track formations are located in the traffic area of public roads; they are separated from the rest of the traffic area by at least kerbstones, hedges, rows of trees, or other fixed physical obstacles [38]. This allows tramways to avoid the traffic jams of road traffic. Segregated bus lanes are non-existing in Braunschweig, except in the area of some tramway stops, where the segregated track formations are used for stopping.

#### 5.1.2. Travel Speed

As already mentioned, the travel speed depends on the infrastructure and vehicle parameters—e.g., acceleration. Factors such as the traffic volume, accessibility, the number of passengers boarding and alighting, and connections at stops also influence the travel speed.

The average speed of the bus lines in Tampere is between 15 and 25 km/h. The new tramway will operate with an average speed of 19–22 km/h and has a maximum speed of 70 km/h [39]. In Tampere, each part of the public transport authority's area has been given a public transport level of service grade from 1 (lowest) to 6 (highest). In grade 6 areas, the travel speed, including the time to reach the stops, should be comparable with PMV [7].

The average speed of the tramway in Germany is 15 up to 20 km/h [40]. In Braunschweig, the tramway has an average speed of between 17 and 19 km/h, depending on the line [25]. The average speed for busses can be assumed to 15 km/h in downtown and 20 km/h for the connection between the suburbs to downtown [40]. Busses for rural areas connect mid-sized and small urban centres with regional centres. These busses achieve an average speed of 20 to 40 km/h [40]. The mentioned average values for busses are similar for Braunschweig.

### 5.1.3. Service Frequency

As shown before, the service frequency depends on the different modes of transport, the density of the surrounding areas, and the time of day. It is mostly determined by economic reasons.

The inner city and major suburbs of Tampere belong to areas with grade 6, in which the service frequency is less than 10 min during rush hour and less than 15 min during evenings and weekends, with a less than 400 m walking distance to stops. Other major areas typically belong to at least grade 4, in which the service frequency is less than 20 min during rush hour and less than 30 min during evenings and weekends, with a less than 800 m walking distance to stops [7].

In Braunschweig, the frequency of a tramway is normally 10, 15, or 20 min, with an average trip length of 4.5 km. The stop distances in the inner city are between 300 and 500 m; in the suburbs, distances expand up to 500 to 600 m [40].

In Braunschweig, the BSVG is also responsible for the operation of all bus lines in the city area. The urban lines run at 10, 15, or 20 min intervals depending on the line, time, or day of the week, with stop distances between 300 to 500 m [40]. A guide value for the maximum distance to the next stop is a walk of 5 min. Catchment areas for stops are not defined, therefore circular buffers are used. This corresponds to a radius of approx. 300 m in Germany [41]. The Finnish Transport Authority suggests that walking distance is 1.3 times the straight distance. For this reason, the radius of the stop buffer is also set to 300 metres [42].

The city bus in Germany is used for trips with an average trip length of 4 km. The frequency of buses for rural areas is typically scheduled to be every 30, 60, or 120 min, and is thus lower than for city busses. The average travel distance is approx. 20 km [40]. These average values are similar for Braunschweig. In Table 2, the most important facts for speed, frequency, and catchment area are presented for comparison.

**Table 2.** Simplified summary for speed, frequency, and catchment area.

|  | Tampere | | Braunschweig | |
|---|---|---|---|---|
|  | **City** | **Suburbs** | **City** | **Suburbs** |
| Avg. speed tramway | 19–22 km/h | - | 17–19 km/h | - |
| Avg. speed bus | 15–25 km/h | 15–25 km/h | 15 km/h | >20 km/h |
| Frequency Tramway (day) | 10–15 min | - | 10–20 min | - |
| Frequency Bus (day) | 10–15 min | >20–30 min | 10–20 min | 20–120 min |
| Catchment Area | 400 m | <800 m | 300–500 m | 500–600 m |

### *5.2. Pricing*

As described in chapter 2, for many people the ticket prices for public transport seems too expensive. The costs of public transport are the second biggest obstacle besides the general preference for the car [6].

### 5.2.1. Price of the Ticket

In Table 3, there are the prices for adult tickets in Tampere and Braunschweig announced for January 2019. Nysse offers a monthly ticket for the Tampere region (2 zones) for 51 € in advance. A monthly ticket in Braunschweig (city) costs 68.50 € [25,43].

**Table 3.** Adult ticket chart announced for January 2019 [25,43].

| Tampere | | | | Braunschweig | | |
|---|---|---|---|---|---|---|
| **Adults** | **30 Days** | **Value Ticket** | **Cash** | **Adults** | **30 Days** | **Single Trip** |
| 2 zones | 51 € | 1.98 € | 3.50 € | City fare | 68.50 € | 2.60 € |
| 3 zones | 69 € | 3.15 € | 5.50 € | Fare level 1 | 70.70 € | 2.80 € |
| 4 zones | 79 € | 4.50 € | 7.50 € | Fare level 2 | 83.90 € | 4.10 € |
| 5 zones | 99 € | 5.85 € | 9.50 € | Fare level 3 | 112.60 € | 5.60 € |
| 6 zones | 109 € | 7.20 € | 11.50 € | Fare level 4 | 154.70 € | 9.00 € |

In Tampere, the introduction of reduced fares led to an increase in the number of passengers in the Nysse region [23]. In Braunschweig, there are no short distance tickets (one to three stops) available. Therefore, passengers have to pay the same price even if their trip is only one station [25].

### 5.2.2. Ticket Selling Network

The way to buy a ticket and its complexity play an important role in choosing the type of public transport. The time required must be minimal and the ticketing system should be simple. There are different types of ticket sales that all have advantages and disadvantages.

In Tampere, single tickets can be purchased from the bus driver, at customer service, and a mobile ticket is also available via the Nysse Mobiilli App. In Tampere, there is also a value ticket via the travelcard available, which provides a price reduction [43].

In Braunschweig, tickets are available at the BSVG ticket agency. Otherwise, a ticket can be purchased directly from the driver in any bus. The tramways will gradually be equipped with ticket machines. Furthermore, you can buy online tickets in Braunschweig via the app of the BSVG [25].

### 5.2.3. Complexity of the Fare System

Nysse offers four types of tickets. The ticket is valid for 1–2 h of travelling, depending on the tariff zone. The tariff zones are divided into zones A to F [43]. The fare is charged in accordance with the number of tariff zones that need to be passed for the trip. A season ticket for the Travelcard is also available in Nysse. The period of validity is 30 or 360 consecutive days. A day ticket covers an unlimited number of journeys in the ABC or ABCDEF zone areas [43]. Nysse has a simple fare-system which mostly divides between monthly or single tickets and the payment option.

In Braunschweig, there are single, 2-pack, and 10-pack tickets for adults and children (6–14 years) offered by the VRB (Verkehrsverbund Region Braunschweig) for all price levels. Buying the 2- or 10-pack tickets reduces the price per trip. When purchasing a day ticket, customers can choose between a group ticket for up to five people and a single day ticket. Additionally, the VRB region is divided into four different tariff zones. There are complex price mechanics for the usage of different tariff zones [44].

These examples show the complexity of the fare-system in Braunschweig. The customer in Braunschweig has to choose from different options, which may lead to refusal of public transport.

### 5.3. Information and Marketing

Marketing and information are important aspects to bring people closer to public transport and to make them aware of it. In the following, the current status of different systems for customer information and brand appearance will be investigated for Tampere and Braunschweig.

### 5.3.1. Onboard Information

As mentioned before, information about the next stops, route, and connections should be clearly visible in the vehicles on displays and also should be announced by voice.

In Tampere, information about the next stops or transfer connections is currently being implemented on screens in all buses. The real-time information covers the majority of the network, which makes the trip more predictable and attractive [43]. All the information is also available in real time in the Nysse app. Furthermore, almost all stops in the inner city are equipped with real-time information.

In newer busses and tramway cars in Braunschweig, there are displays on board. These show the progress of the route and the next stop. Different connections at the respective stations are shown symbolically. In older vehicles (bus and tramway cars), there is a display showing only the next station, but it also happens that there are no displays or announcements about the next station.

### 5.3.2. Information at Stops

As already mentioned, the city centre of Tampere has real-time information available on every bus stop in inner city area. On the other hand, stops in rural areas and some suburbs of Tampere are often only identifiable by a stop sign. In addition to the lack of accessibility, the passengers do not have the opportunity to obtain analogue information about fares or departures. Simplified representations provide benefits for people who do not have access to mobile devices. Timetable displays and route network maps are available at most stops, but these are sometimes small and hard to understand for unexperienced public transport users.

In Braunschweig, there are only digital displays with the live departure times at the busiest stations inside the city. However, the equipment of bus stations in Braunschweig offer more information than in Tampere. Each bus stop in Braunschweig provides paper timetable boards with information about travel times and fares, while the finish stops lack analogue information.

### 5.3.3. Digital Information

Digital information is necessary for a better developed service and a barrier-free access to multi-modal and intermodal connections across all modes of transport.

For operational reasons and information sharing with the customers, buses in Tampere are equipped with satellite navigation receivers. By processing this information and transmitting it to the control centre, the computer compares actual and target values in order to process real-time information; third parties can obtain this information free of charge via the open access interface Journeys API. The Tampere City Council publishes a general transit feed specification (GTFS) on the Tampere website, which contains real-time information as well as all types of transit data such as timetables, bus stops, and journeys. Nysse transmits these data to the displays at the stops, and even on their smartphone application. Passengers can see immediately if the bus is late and long waiting times at bus stops can be avoided [43].

Real-time information with GTFS data does not exist in Braunschweig. The information system in BSVG buses will be converted from radio to GSM in this year. With this innovation, network-wide in-house monitoring will be possible without interruption for the first time in comparison to the radio communication system used today [36].

### 5.3.4. Brand Appearance

The brand appearance is important to make the public transport or the means of transport attractive for new customers and present a modern image.

In Tampere, the Nysse brand is used in all buses regardless of the operator. Buses are white and blue, with the Nysse logo and text clearly visible. The brand is also used on all printed materials, such as the bus stop information and all online content and apps.

BSVG's brand identity is very uniform in the city. Advertisements are made at almost all stops of the BSVG. The BSVG is not advertised in surrounding cities in the Braunschweig area in order not to be a competitor [36].

*5.4. Quality of Travel*

The quality of travel comfort in public transport can be differentiated in three categories: ease of boarding and alighting, seating capacity, as well as the safety and equipment at stops. The quality of travel is seen as an obstacle by many people to change from their own car to public transport.

5.4.1. Ease of Boarding and Alighting

The quality of the transportation process itself highly depends on the vehicle. Both cities have vehicles that are designed for peak times and transporting pupils. It should also be mentioned that every bus in Braunschweig and Tampere is low-floored and equipped with fold-in ramps to ensure accessibility for people with disabilities, luggage, or strollers. A total of 90% of BSVG's tramway is low-floor trains, and therefore it is almost completely barrier-free [28].

5.4.2. Seating Capacity

The seating capacity is an important characteristic to evaluate the attractiveness of the public transport. In Tampere, buses from Solaris are mainly equipped with approx. 37 to 54 seats depending on the bus model (Urbino 12 or Urbino 18) [45]. The seating capacity of the individual means of transport in Braunschweig varies. Busses have around 50 seats and standing room for the same amount of people, while the tramway cars have nearly double the space (90 seats, 121 standing places) [28,46]. The capacity utilisation of tramways and buses varies according to the time of day. In the peak hours, there are a lot of passengers in the vehicles and not every passenger gets a seat, which is acceptable with an average travel distance of 4 km. In Germany, the capacity is designed for peak hours. The occupancy rate of seats can be exceeded over 100%, but the total capacity of seating and standing should be utilised to a maximum of 65% [46].

5.4.3. Safety and Equipment at Stops

The equipment of a stop is important for the passenger's feeling of safety. If a stop is unlit and only recognisable by a stop sign, this is insufficient for comfort and safety.

In Tampere, there are bus stops that can only be identified by a bus stop sign, but do not have a bus stop bay or other equipment. In addition to that, many stops are not barrier-free and do not provide adequate seating or weather protection.

In Braunschweig, all of the BSVG tramway stations are designed for low-floor use, with the exception of four tramway stops. Further requirements for the tramway stops are barrier-free access for people with limited mobility, lighting, dynamic passenger information system (according to the project "Echtzeit" (real time)), timetable display, clock, seating and weather protection, etc. In recent years, the municipalities have already carried out numerous modernisations and improvements at bus stops, but the infrastructure and thus the quality of the infrastructure still varies greatly. Most bus stops are accessible for persons with reduced mobility [47].

In summary, the public transport in Tampere and Braunschweig is well developed but still has some weaknesses. The following summarises the most important aspects from chapter 5.

In Tampere, the level of service is quite good, with its real-time information, high travel speed, and service frequency. The prices, ticket sales, and the complexity of the fare system are fair and easy to understand in Tampere.

The general level of service in Braunschweig is quite good as in Tampere, especially with the tramway, which run most of the time on separated track formations, but, e.g., there is no real time information available in the bus and tramway. In Braunschweig, the pricing for public transport users is customer unfriendly. There are different zones and various ticket types. Furthermore, the ticket via the BSVG app does not always work properly.

In conclusion, the information section shows that the public transport in Tampere is far more digitised than in Braunschweig. In terms of marketing, Tampere is more present with a more uniform

look than in the Braunschweig region. The quality of travel in both cities according to the accessibility and seating capacity is good, except for some minor aspects, such as equipment at stops.

## 6. Future Improvements

In the following and based on the analysis of quality factors in Tampere and Braunschweig, aspects will be described which should improve public transport in both cities. Specific improvements are based on the investigation of different level of service attributes. The following focus is on the most useful and cost-effective actions for future improvements. Possible actions to promote public transport and to increase its attractiveness are given. Best practise examples from other cities are considered and explained in the context.

### 6.1. Marketing for Public Transport

The need for marketing measures for public transport in Braunschweig is pointed out in its long-term planning as an important topic [47]. Especially in rural areas, an intense promotion is necessary to reach a satisfactory number of passengers. To win new customers, there should be targeted publication of offers in many information channels, such as online, in newspapers, via radio, etc. Another idea is advertising by discounted fares, like was done in Helsinki in 2013. A total of 28,000 2-week tickets were given to new customers to experience public transport on a trial basis. Approximately a third of them continued their use and the ticket sales quadrupled [23,48].

Another approach in the long-term planning of Braunschweig is the perception of the entire public transport as one system [47]. Therefore, all bus stops should have uniform timetables and stop designs. Tampere also has a lot to improve on these aspects. BSVG has already revised its brand image in summer 2019 [48]. Besides this, the homepage and the mobile apps should get an update to be more appealing to customers. It should be mentioned that the Nysse homepage is also available in English, while the BSVG homepage is only available in the German language.

### 6.2. Digital Fare Models and Better Information

In Braunschweig, only a small amount of public transport users use digital tickets, because the available app and website are out-of-date. Mobile tickets should be extended for the entire area of the transport association. Furthermore, the integration of local rail transport should be included in the normal tariff zones; this is already the case in the Braunschweig region [44]. To buy a ticket in the BSVG application, you have to register first. The application itself is not functioning very well and seems outdated [49]. In Tampere, Nysse also gives you the option to buy a ticket without prior registration. An environmental ticket could also be a way to increase the number of users. One option is to connect the ticket to a Park and Ride System (P&R). Advantages must be offered—for example, a discount fare—or the parking costs in the city centre must be higher.

As already mentioned, the transport systems will be switched from normal radio to GSM in Braunschweig. This will ensure that real-time data will be available in vehicles, at stops, and also via the app in the course of the next year.

### 6.3. Best Price Guarantee

Since the tariff systems in both cities are complex, it is advisable to introduce a best prise guarantee. As an example, the Oyster Card in London and also the KolibriCard in Schwäbisch Hall Germany determine the cheapest travel cost by always choosing the cheapest fare in the first case. The system in London never charges more than the cost of the day ticket [50,51]. This system could be implemented in Tampere due to the already existing smartcard. It is one hurdle less, as the customer does not have to deal with the difficulties of ticket prices and tariff zones.

### 6.4. Integration in MaaS

Mobility as a Service (MaaS) has its origin in Helsinki. MaaS has already been part of the local transport for several years in Tampere. It describes a relocation away from individual modes of transport and towards mobility solutions that are consumed as a service. The service is able to combine transportation services from public and private transportation providers to enable people using a multimodal system finding the best solution for every transportation need. For example, it combines traditional public transport, such as busses or the tramway, with bike-/ride-/scooter-sharing services [52].

In Helsinki, a successful service called Whim offers mobility services for fixed monthly rates and combines it even with car rental options (Whim). While Finland is ahead in digitalism, there are a couple of MaaS apps starting slowly in Germany. Like Moovel [53] or Jelbi [54], their coverage of cities and multimodal services is limited. One big issue in Germany is the available data and the cooperation between city-owned public transport services and new app-driven businesses.

### 6.5. Demand Responsive Transport Services

In Germany, there are various alternative and flexible forms of operation. There are on-demand transport services which operate after prior reservation by telephone. Demand controls the number of trips and, depending on the type of on-demand traffic, the route. The on-demand operation can be divided into four different types: line operation, directional band operation, sector operation, and area operation. The different operation modes can be combined with each other.

In the region of Tampere, the service PALI (abbr. Palveluliikenne (service transport)) provides mobility for aged persons as well as for people in wheelchairs or others with limited mobility. This service could be extended to include demand-driven transport to pro-vide a direct link to the nearest railway stations or to main and regional centres [55].

Alternative and flexible operating modes are particularly suitable for rural areas and are-as with low demand. In summary, this means non-regulated transport services lead not only to the substitution of public transport but also to an environmental impact. For this reason, on-demand integrations must always take place in harmony with public transport and the local public transport authorities.

### 6.6. Level of Service

The main aspects for choosing public transport are the reliability on the system and its transfers offered. This mainly focusses on the on-time performance, as well as travel speed and service frequency. Braunschweig would benefit from implementing a similar level of service grading to the areas of the city as Tampere is using. Both cities rely on expanding their tramway network in the future, which considerably improves the level of service on the routes where buses are replaced with tramway, but may also allow improvements in the level of service elsewhere due to the cost-efficient operation of the tramway compared to buses. More information is explained at [23,24,56].

## 7. Conclusions

Mobility is an important aspect of everyday life. Rising car numbers and congested traffic results in noise, air pollution, and stressed street users. To switch people from using their own car to using public transport instead, public transport needs to be attractive and fulfil the needs of its potential users.

The aim of this study was to (1) explore the level of service attributes of public transport in Tampere and Braunschweig and evaluate their current state and (2) give advice for each city to further increase the ridership

Regarding the first aim of the study, both public transport systems have been gaining new users over the last 20 years, so it is obvious that they are doing well regarding some of their chosen attributes. However, regarding the second aim of the study, there were some aspects identified to increase the attractiveness of both public transport systems. One of the mayor points is the need for digitalization

in Germany, where most mobile applications, websites, and especially user behaviour seem to be out-of-date. Another aspect that can be changed by the local public transport companies is their marketing and information policy. They both lack advertising their own system, which is necessary to get people to change from using their private car to using public transport vehicles. Important for the change is also the quality of vehicles and available seating capacity, as well as easy-to-understand and affordable pricing.

For those aspects, possible improvements are given in this paper and supplemented with examples from other use cases. One key aspect which needs to be improved is the cooperation between different mobility services to create and implement a Mobility as a Service system. This could make the user experience much easier. Current MaaS providers, e.g., in Helsinki, are working on getting people to using public transport. It is possible to use MaaS in rural areas, where demand-responsive public transport is the most economical way to offer public transport.

The main aspects for choosing public transport are the reliability on the system and its transfers offered. This mainly focusses on the on-time performance, as well as the travel speed and service frequency. Those aspects can be fulfilled with better infrastructure and vehicles. Tampere and Braunschweig are both working on expanding their tramway systems to reduce travel times and increase on-time performance as well as seating capacity. The tramway improvements are the key in both cities maintaining their increase in public transport ridership in the future.

**Author Contributions:** Conceptualization, C.S., N.S., H.L., T.S.; methodology, C.S., N.S., H.L.; validation, C.S., N.S., H.L., T.S.; formal analysis, C.S., N.S., H.L.; investigation, C.S., N.S., H.L.; resources, C.S., N.S., H.L.; writing—original draft preparation, C.S., N.S., H.L.; writing—review and editing, C.S., N.S., H.L., T.S.; visualization, C.S., N.S.; supervision, H.L., T.S.; project administration, H.L.,T.S.; funding acquisition, H.L., T.S. All authors have read and agreed to the published version of the manuscript.

**Funding:** This work was supported by the Joint call for proposals to enhance collaboration between Tampere University of Technology and Technische Universität Braunschweig.

**Conflicts of Interest:** The authors declare no conflict of interest.

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
