# Peer review of "How to Achieve a Continuous Increase in Public Transport Ridership?—A Case Study of Braunschweig and Tampere"

_sustainability, doi:10.3390/su12198063_

Round 1

Reviewer 1 Report

This manuscript is devoted to study the service level of public transport in Tampere and Braunschweig and to give advice for each city to increase the ridership.
"Introduction" is enough for problem understanding.
"Theoretical background" has too many references to data analytics and doesn't provide references to similar studies.
"Methods used" doesn't contain reliable descriptions regarding the methods.
"Results" are commented enough but what are results?

The study design is unclear.

The manuscript is badly formatted.
Suggestion for formatting improvement:
1. Are "PT" and "LR" acronyms using really need? Full "Public transport" and "light rail" are quite short.
2. Table 1 is too wide and has no caption.
3. Table 2 is not a table, it looks like a Figure. And there is no caption.
4. Table 3 has no caption.
!3. Lines 219-230 contain some text from the article template.
4. The heading of this article has some mistakes in numeration. For example, Head "3. Methods used" and "3. PT systems in Tampere and Braunschweg" have the same number; at the line 296 Head "5.1.1 Level of Service" has the wrong number, and so on.

English proofreading is required.

Reviewer 2 Report

Table 1 is difficult to understand, good quality of presentation is missing, please provide bibliographic entries in square brackets [..].

lines 219-230, please delete the sample text from MDPI template

please renumber the chapters starting with PT systems in Tampere and Braunchweg (should be 4)

I believe that the term light rail is not a correct term for a tram. Please use the terms tram for conventional tram systems (in US streetcar is a proper name for conventional tram, and the light rail is a fast tram or tram-train).

Analyzing the presented article, I do not see a real assessment of the transport system. I see an encyclopedic description of two cities. There is no clear indication of which aspects can predict travel growth and which are barriers to growth. There is no indication of the methodology of the evaluation and its performance. There is no summary of the existing state, no indication of strengths and weaknesses. Only on this basis can future solutions be identified.

Meanwhile, the chapter 5.2 "future improvements" is intertwined with the analysis of the weaknesses of the current state with recommendations for the future. There is no clear structure of the text. Please work on this aspect. The future improvements schould be in a separate chapter, which should be a discussion of the results from chapter 5.

It is worth giving a summary of chapter 5 in a graphic form or in the form of a percentage in the table... So that the reader does not have to remember 5 pages of the text.

Reviewer 3 Report

This paper presents the status quo of public transport (PT) in two selected cities in Germany and Finland. The main criteria for the PT evaluation are presented according to the literature. Some comparisons are given to shown the similarities in PT network and operation. It was mentioned that the two PT networks should be extended but the new systems are not yet in operation thus the benefit of the new systems cannot be evaluated yet. Anyway this paper shows some ideas how to improve the acceptance of PT in the European countries.

Some detailed comments:

It is desirable to present the key parameters of the attributes in tables for a better comparison.

It is desirable to define network-wide indexes for the parameters of the corresponding attributes.

Round 2

Reviewer 1 Report

The authors provided a new version of the manuscript and took into account all comments that I make, so I think this version could be accepted in the present form.

Author Response

Thank you for your feedback and comments on our paper.

Reviewer 2 Report

There was no version with track changes to quickly evaluate the changes made.

The term light rail was left in line 317,

It is still difficult to determine why exactly the improvements mentioned in chapter 6 have been selected. The reason is the lack of a clear purpose and summary of chapter 5. The tables added do not sum up the chapter. Since improvements are being made, why do you not provide pre- and post-status assessments?

Author Response

Thank you for your feedback and comments on our paper.

We have made the changes in lines 282 to 284 and in lines 488 to 494 to clarify the contents of chapters 5 and 6.

Our aim of the paper is to determine which aspects can lead to higher passenger numbers that's the reason why we don't provide pre- and post-status assessments.

We changed the word light rail in line 320 to tramway.

This time, we use the track changes feature.

Reviewer 3 Report

My suggestions in previous comments are addressed properly.
Thus, I don't have any further comments. I can now recommend a publication.

Author Response

Thank your for your feedback and comments on our paper.